# Rhosin Suppressed Tumor Cell Metastasis through Inhibition of Rho/YAP Pathway and Expression of RHAMM and CXCR4 in Melanoma and Breast Cancer Cells

**DOI:** 10.3390/biomedicines9010035

**Published:** 2021-01-04

**Authors:** Masanobu Tsubaki, Shuuji Genno, Tomoya Takeda, Takuya Matsuda, Naoto Kimura, Yuuma Yamashita, Yuusuke Morii, Kazunori Shimomura, Shozo Nishida

**Affiliations:** 1Division of Pharmacotherapy, Faculty of Pharmacy, Kindai University, Kowakae, Higashi-Osaka 577-8502, Japan; tsubaki@phar.kindai.ac.jp (M.T.); genno_kindai@yahoo.co.jp (S.G.); takeda@phar.kindai.ac.jp (T.T.); takuya.matsuda.kindai@gmail.com (T.M.); kimura_naoto_kindai@yahoo.co.jp (N.K.); yamashita_kindai@yahoo.co.jp (Y.Y.); morii_kindai@yahoo.co.jp (Y.M.); 2Department of Phamacy, Municipal Ikeda Hospital, Ikeda, Osaka 563-0025, Japan; shimomura_ikeda@yahoo.co.jp

**Keywords:** Rho, rhosin, YAP, breast cancer, melanoma, metastasis, RHAMM, CXCR4

## Abstract

The high mortality rate of cancer is strongly correlated with the development of distant metastases at secondary sites. Although Rho GTPases, such as RhoA, RhoB, RhoC, and RhoE, promote tumor metastasis, the main roles of Rho GTPases remain unidentified. It is also unclear whether rhosin, a Rho inhibitor, acts by suppressing metastasis by a downstream inhibition of Rho. In this study, we investigated this mechanism of metastasis in highly metastatic melanoma and breast cancer cells, and the mechanism of inhibition of metastasis by rhosin. We found that rhosin suppressed the RhoA and RhoC activation, the nuclear localization of YAP, but did not affect ERK1/2, Akt, or NF-κB activation in the highly metastatic cell lines B16BL6 and 4T1. High expression of YAP was associated with poor overall and recurrence-free survival in patients with breast cancer or melanoma. Treatment with rhosin inhibited lung metastasis in vivo. Moreover, rhosin inhibited tumor cell adhesion to the extracellular matrix via suppression of RHAMM expression, and inhibited SDF-1-induced cell migration and invasion by decreasing CXCR4 expression in B16BL6 and 4T1 cells. These results suggest that the inhibition of RhoA/C-YAP pathway by rhosin could be an extremely useful therapeutic approach in patients with melanoma and breast cancer.

## 1. Introduction

Tumor metastasis is defined by the spread of cells from the primary tumor to new tumor colonies in distant tissues; it causes more than 90% of cancer-related deaths [1]. The tumor metastatic process is comprised of multiple steps: tumor cell invasion, migration, and adhesion [2]. Tumor cells locally invade the surrounding tissues of the primary lesion, the microvasculature of the lymph and blood vessels, and migrate through the blood flow to the microvessels of the distant tissues, such as the lung, bone, and liver. They adhere to microvessels around new tissues, exit from the vessels and, finally, form a secondary tumor [2]. Despite the development of therapies to combat primary tumors, the mortality rate of metastatic tumors still remains high [3]. Therefore, new molecular therapeutic approaches are essential to combat tumor metastasis.

Malignant melanoma accounts for 4% of all cancers, is the most aggressive of all skin cancers and accounts for 80% of death among skin cancers [4]. Although early diagnosis of melanoma can be surgical removal, stage IV metastatic melanoma patients have a five-year survival rate of 18% [5]. Recently, the development of molecular targeting drugs such as BRAF inhibitors, mitogen-activated protein kinase kinase 1/2 (MEK1/2) inhibitors, anti-programmed cell death protein 1 (PD-1) antibodies, and anti-cytotoxic T-lymphocyte associated protein 4 (CTLA-4) antibodies, improved the overall survival rate [6,7]. However, anti-PD-1 antibodies and anti-CTLA-4 antibodies have shown an overall response rate of 16.7% and 0% to 16.7%, respectively, and patients with malignant melanoma have developed early resistance or acquired resistance to these drugs [8,9]. Thus, other molecular therapeutic approaches are essential for treating metastatic melanomas.

Breast cancer is the most common cancer in women worldwide, and early diagnosis of breast cancer can be curable in 70–80% of patients [10]. However, the development of metastatic breast cancer is poor prognosis and accounts for over 90% of breast cancer-related mortality [11]. Breast cancer patients with metastasis accounts for 21–32% in the lung, and lung metastases are still difficult to treat, and it is estimated that 60–70% of patients who die from breast cancer have lung metastases [11]. Therefore, understanding the mechanisms that facilitate lung metastasis is of importance.

RhoGTPases, such as RhoA, RhoB, RhoC, and RhoE, play an important role in reorganizing the actin cytoskeleton, cell migration, proliferation, polarity, and vesicle trafficking [12]. It has been shown that RhoA and RhoC are overexpressed in various cancers, including breast cancer, lung cancer, and melanoma, and are correlated with tumor metastasis [13,14]. RhoB is generally known to be a tumor suppressor gene, since RhoB inhibits cell proliferation, invasion, and tumor angiogenesis, and overexpression of RhoB induces apoptosis in colorectal carcinoma cells [15]. However, RhoB expression is correlated with glioblastoma tumorigenesis via suppression of p53 and p21, and overexpression of RhoB in breast cancer is involved in the progression of the disease [16,17]. In addition, RhoE accelerates cell migration, invasion, and metastasis via increased CXC chemokine receptor 4 (CXCR4) expression in gastric cancer cells, and expression of RhoE in melanoma cells is associated with invasive behavior in a three-dimensional dermal-like environment [18,19]. However, various reports have indicated a tumor suppressor role for RhoE in tumor growth and metastasis [20]. Thus, modulation of the activation/expression of Rho GTPases may provide a molecular mechanism for anti-tumor metastasis therapy.

Activation of Rho GTPases induces the phosphorylation of Rho-associated coiled-coil-containing protein kinase (ROCK), which acts as an upstream regulator activating the extracellular signal-regulated kinase 1/2 (ERK1/2), nuclear factor κB (NF-κB), Akt, and yes-associated protein (YAP) [21,22,23]. These downstream molecules induce the transcription of genes including chemotaxis-related factors, such as CC chemokine receptor 7 (CCR7) and CXCR4, and cell adhesion molecules, such as hyaluronan receptor CD44 and receptor for hyaluronan-mediated motility (RHAMM) [24,25,26,27]. In addition, CCR7, CXCR4, CD44, and RHAMM bind to these ligands such as chemokine ligand 19 (CCL19), CCL21, stromal cell-derived factor 1 (SDF-1), hyaluronic acid, type I collagen, type IV collagen, and fibronectin, which activates the Rho/ROCK, phosphoinositide 3-kinase (PI3K)/Akt, and NF-κB pathways [28,29,30]. However, the main roles of Rho GTPases and Rho GTPase downstream signaling molecules in tumor metastasis remain unidentified.

Rhosin is known to bind to trp58 of RhoA and inhibit its activation by Rho guanine nucleotide exchange factor, and to inhibit the activation of RhoC [31]. In addition, rhosin inhibited cell migration and invasion in the human pancreas cancer cell line PANC-1 [32]. However, it remains unclear whether rhosin acts by inhibiting metastasis by blocking the Rho downstream signaling. In the present study, we evaluated the main roles of Rho GTPases that are correlated with tumor metastasis by using highly metastatic melanoma and breast cancer cells, and investigated the mechanism of inhibition of metastasis by rhosin.

## 2. Materials and Methods

### 2.1. Cell Culture 

B16F1 and B16BL6 cells were supplied by Dr. Inufusa (Kindai University, Osaka, Japan). B16F1 cells were established from lung metastasis lesion of intravenous injection of B16 melanoma cells into a syngeneic C57BL/6J mouse, and B16F10 cells were selected by 10 successive lung metastasis lesions. B16BL6 cells were derived from B16F10 cells that have invaded the mouse bladder membrane [33]. 4T1 cells were provided by the American Type Culture Collection (Rockville, MD, USA). We used Balb/c-derived 4T1 cells because 4T1 cells and breast cancer have similar metastatic sites [34]. These cells were maintained in RPMI1640 medium (Sigma, St Louis, MO, USA) containing 10% fetal bovine serum (FBS) (Gibco, Carlsbad, CA, USA), 100 μg/mL of penicillin (Gibco), 100 U/mL of streptomycin (Gibco), and 25 mM 4-(2-hydroxyethyl)-1-piperazineethanesulfonic acid (pH 7.4; FUJIFILM Wako, Tokyo, Japan) in an atmosphere containing 5% CO_2_.

### 2.2. Western Blotting

The cells were lysed with ProteoExtract Subcellular Proteome Extraction kit (Calbiochem, San Diego, CA, USA) and extracted cytoplasm and nuclear fraction following the manufacturer’s instructions, and the cell lysates were separated on SDS-polyacrylamide gels and blotted onto PVDF membranes (GE Healthcare, Buckinghamshire, UK).The membranes were reacted overnight at 4 °C with each primary antibody: anti-phospho-ERK1/2 (#9101), anti-ERK1/2 (#9102), anti-phospho-Akt (#9271), anti-Akt (#9272), anti-phospho-NF-κB p65 (#3031), anti-NF-κB p65 (#3034), anti-YAP (#8418) (Cell Signaling Technology, Beverly, MA, USA), anti-RhoA (F-1), anti-RhoB (C-5), anti-RhoC (C-16), anti-RhoE (clone 4), anti-RHAMM (C-9), anti-CD44 (F-4), anti-CXCR4 (4G10), anti-CCR7 (A-19), lamin A/C antibody (H-110) (Santa Cruz Biotechnologies, CA, USA), and anti-β-actin antibody (clone AC-74) (Sigma). The membranes were reacted with horseradish peroxidase-coupled sheep anti-rabbit IgG (GE Healthcare) for 1 h and were subsequently visualized using Luminata Forte (Merck Millipore, Nottingham, UK).

### 2.3. Rho Pull-Down Assay

Rho pull-down assay was performed as previously described [24].

### 2.4. In Vitro Migration, Invasion, and Adhesion Assays

Cell migration and invasion assay was performed as previously described [35]. The cells were seeded into the upper chambers in FBS-free RPMI1640 medium after adding rhosin (Merck Millipore) or verteporfin (Cayman Chemical, Ann Arbor, MI, USA). The lower chambers were loaded with FBS-free RPMI1640 medium including 50 ng/mL SDF-1 (PeproTech, London, UK). The cell adhesion assay was performed as previously described [35].

### 2.5. RNA Interference

The Silencer^®^ select small interfering RNAs (siRNAs) for RhoA and RhoC were obtained from Invitrogen (Carlsbad, CA, USA). Silencer^®^ select negative control (Invitrogen) was used as a negative control. Transfection of siRNAs was performed according to the manufacturer’s protocol by using the LipofectAMINETM 3000 reagent (Invitrogen). 

### 2.6. Mice

Female Balb/c mice (age, 8 weeks) and female C57BL/6J mice (age, 8 weeks) were obtained from the Shimizu Laboratory Animals (Kyoto, Japan). The mice were housed in specific pathogen-free conditions. All animal experiments were conducted according to the guidelines of the United Kingdom Coordinating Committee on Cancer Research (2010) and Animal Care and Use Committee of the Kindai University (project identification code KAPS-27-021, 1 April 2015).

### 2.7. Effects of Intraperitoneal Administration of Rhosin on Lung Metastasis of Tumor Cells

1 × 10^5^ B16BL6 and 4T1 cells were administered into the tail vein of C57BL/6J mice and Balb/c mice. B16BL6- and 4T1-bearing mice were randomly divided into three groups, and were treated with 0.1% dimethyl sulfoxide (control), and 10 mg/kg or 30 mg/kg of rhosin for 14 days. After 14 days, the mice were sacrificed, and the lung tissues of each mouse were excised. The tissues were fixed in a neutral-buffered formaldehyde solution, and the lung nodules were counted as black or white forms.

### 2.8. SurvExpress Analysis

The relationship between the overall survival of melanoma or breast cancer patients and YAP expression was evaluated using the SurvExpress database [36]. The microarray datasets of Schmidt, Loi, and Jönsson were assessed using the SurvExpress database [37,38,39].

### 2.9. Statistical Analysis

All results were represented as the mean ± SD of several independent experiments. Analyses were conducted using SPSS version 21.0 software (IBM Inc., Chicago, IL, USA), and Shapiro–Wilk analysis and one-way analysis of variance (ANOVA) were performed. When no differences on Shapiro–Wilk and satisfactory differences on ANOVA were confirmed, the control group and various drug-treated groups were compared and analyzed using Dunnett’s test. *p*-values of < 0.05 were considered as significant.

## 3. Results

### 3.1. Overexpression and Activation of RhoA and RhoC, and Their Downstream Signaling Molecules Contribute to High Metastatic Potential

Our previous study has shown that RhoA/C overexpression was involved with poor prognosis in patients with malignant melanoma [35]. In addition, we investigated whether RhoA and RhoC expression contributes to poor prognosis in patients with breast cancer. Patients with high expression of RhoA and RhoC had shorter overall survivals than patients with low expression of RhoA and RhoC, and high expression of RhoC correlated with short recurrence-free survival, but this was not the case for high expression of RhoA (Appendix A). Thus, overexpression of RhoA and RhoC is potentially involved in metastasis formation, and patient relapse and mortality rates. We also examined the relationship between the expression and activation of Rho family proteins and high metastatic potential. RhoA and RhoC expression in B16BL6 cells (cell line with high metastatic potential) was higher than in B16F1 cells (cell line with low metastatic potential) (Appendix A), but this was not the case for RhoB and RhoE. In addition, the activation of RhoA and RhoC in B16BL6 cells was increased compared to that in B16F1 cells, but activation of RhoB was enhanced in B16F1 cells than B16BL6 cells (Figure 1a). Moreover, we confirmed that 4T1 cells, a highly metastatic mouse breast cancer cell line [40,41], expressed total and GTP-form RhoA and RhoC (Appendix A).

Next, we investigated the activation of RhoA and RhoC downstream signaling molecules. B16BL6 cells activated ERK1/2, Akt, NF-κB, and YAP proteins (Figure 1b) and these signal molecules’ activation was similarly detected in 4T1 (Appendix A). These results might indicate that overexpression/activation of RhoA and RhoC increased metastasis through ERK1/2, Akt, NF-κB, and YAP activation.

### 3.2. Rhosin Inhibits YAP Activation via Inhibition of RhoA and RhoC

To investigate the cytotoxic effects of rhosin on B16BL6 and 4T1 cells, cell viability was assessed by treating cells with 1–100 μM rhosin. Rhosin at a concentration of 100 μM induced cell death in B16BL6 and 4T1 cells (Appendix A). On the basis of these results, we determined that 1–50 μM rhosin were not cytotoxic to B16BL6 or 4T1 cells.

Next, we examined whether rhosin, a RhoA/C inhibitor, suppressed the downstream signaling molecules of RhoA and RhoC in B16BL6 and 4T1 cells. Rhosin inhibited RhoA and RhoC activation, and rhosin inhibited RhoC more strongly than RhoA. In addition, rhosin suppressed YAP activation in B16BL6 and 4T1 cells, but did not affect ERK1/2, Akt, and NF-κB activation in a concentration-dependent manner (Figure 2a,b). In addition, inhibited expression of RhoA and RhoC by treatment with siRNA suppressed the YAP nuclear translocation and enhanced the cytoplasmic expression of YAP in B16BL6 cells (Figure 2c,d). These results indicated that RhoA and RhoC promote YAP activation, and the YAP pathway may be a major Rho signaling pathway.

We also examined whether YAP expression contributed to poor prognosis in patients with breast cancer and melanoma. YAP-high expression in patients with melanoma had shorter overall survival than YAP-low patients with melanoma (Appendix A). In addition, patients with high expression of YAP had shorter overall and recurrence-free survivals than patients with low YAP expression in breast cancer (Appendix A). Thus, overexpression of YAP is potentially involved in metastasis formation and affects patient relapse and mortality rates.

### 3.3. Inhibitory Effect of Rhosin on Lung Metastasis in Mice Injected with B16BL6 and 4T1 Cells

We also investigated whether rhosin suppressed tumor metastasis in an experimental metastasis model. The number of lung metastatic nodules in B16BL6 and 4T1 cells diminished after administration of rhosin in a dose-dependent manner (Figure 3a). In addition, rhosin suppressed 4T1-luc tumor cell metastasis to the lung region as revealed by the reduction in photon flux (Figure 3b).

### 3.4. B16BL6 Cells Have Increased Expression of RHAMM, CD44 and CXCR4, and Rhosin Suppressed Cell Adhesion to Type I and IV Collagen via Inhibition of RHAMM Expression

Cell surface molecules, such as chemokine receptors and hyaluronan receptors, promote tumor cell migration, invasion, adhesion, and metastasis. CXCR4 and CCR7 are known to promote tumor metastasis via cell migration and invasion [42,43]. RHAMM and CD44 play a role in cell adhesion to the extracellular matrix (ECM) and are required for tumor cell invasion and metastasis [42,44]. Next, we examined the expression of RHAMM, CD44, CXCR4 and CCR7 in B16BL6 and B16F1 cells. We found that B16BL6 cells had higher expression of RHAMM, CD44 and CXCR4 than B16F1 cells, but there was no difference in CCR7 expression among the cell lines (Figure 4a).

Next, we investigated the inhibitory effects of rhosin on the expression of RHAMM and CD44 adhesion molecules to type I and type IV collagen in B16BL6 cells. Rhosin markedly inhibited the expression of RHAMM, but not CD44, in a concentration-dependent manner (Figure 4b). In addition, rhosin significantly suppressed cell adhesion to type I and type IV collagen in B16BL6 cells (Figure 4c). Moreover, rhosin inhibited the cell adhesion to type I and type IV collagen via suppression of RHAMM expression in 4T1 cells (Figure 4d,e). Moreover, inhibited expression of RhoA and RhoC by treatment with siRNA suppressed the RHAMM expression, and cell adhesion to type I and type IV collagen in B16BL6 cells (Figure 4f–h).

### 3.5. Rhosin Suppressed Cell Migration and Invasion via Inhibition of CXCR4 Expression

Cell migration and invasion are important processes in metastasis. We examined the inhibitory effect of rhosin on CXCR4 expression, and tumor cell migration and invasion in B16BL6 cells. Rhosin inhibited CXCR4 expression, and SDF-1-induced cell migration, and invasion in B16BL6 cells (Figure 5a,b). In addition, rhosin suppressed the SDF-1-induced cell migration and invasion via decreasing CXCR4 expression in 4T1 cells (Figure 5c,d). Moreover, inhibited expression of RhoA and RhoC by treatment with siRNA suppressed the CXCR4 expression, and SDF-1-induced cell migration, and invasion in B16BL6 cells (Figure 5e,f).

### 3.6. Verteporfin, a YAP Inhibitor, Suppressed the Cell Adhesion, Migration and Invasion via Inhibition of RHAMM and CXCR4 Expression

Our results suggest that Rhosin suppressed the cell adhesion, migration, invasion and metastasis via inhibition of Rho/YAP pathway activation and expression of RHAMM and CXCR4. We examined whether verteporfin, a YAP inhibitor, inhibits the cell adhesion, migration and invasion via suppression of RHAMM and CXCR4 expression in B16BL6 cells. To investigate the cytotoxic effects of verteporfin on B16BL6 cells, cell viability was assessed by treating cells with 0.1–5 μM verteporfin. Verteporfin, at a concentration of 5 μM, induced cell death in B16BL6 cells (Appendix A). On the basis of these results, we determined that 0.1–2 μM verteporfin were not cytotoxic to B16BL6 cells. In addition, verteporfin inhibited the cell adhesion to type I and type IV collagen via suppression of RHAMM (Figure 6a,b) in concentration-dependent manner. In addition, verteporfin suppressed the SDF-1-induced cell migration and invasion through inhibition of CXCR4 expression in B16BL6 cells (Figure 6c,d).

## 4. Discussion

In the present study, we demonstrated that overexpression and activation of RhoA/C promoted lung metastasis, and increased the relapse and patient mortality rates. It has been reported that activation of RhoA increases in human breast cancer tissues depending on the stage [45]. It has also been shown that the level of RhoA expression correlates with the histopathological degree of cancer, and RhoC expression correlates with the extent of local invasion in colorectal carcinoma patients [46]. In melanoma patients, RhoC expression has been involved with the presence of lymphatic metastasis at the time of diagnosis and shorter disease-free and overall survival rates [47]. Our previous study showed that overexpression of RhoA/C correlates with poor overall survival in melanoma patients [35]. These results suggest that overexpression and activation of RhoA/RhoC are strongly associated with the metastasis, relapse and poor prognosis in patients with melanoma and breast cancer.

RhoA and RhoC induce multiple signaling pathways, including the PI3K/Akt pathway, mitogen-activated protein kinase kinase 1/2 (MEK1/2)/ERK1/2 pathway, and ROCK pathway in various cancer cells [48,49,50]. It has been shown that the Rho pathway activates NF-κB and YAP in cancer cells [51]. We found that the highly metastatic tumor cell lines of B16BL6 had increased levels of phosphorylated ERK1/2, Akt, and NF-κB expression, and induced more nuclear translocation of YAP than B16F1 cells. In addition, inhibition of RhoA/C by rhosin, a RhoA/C inhibitor, increased the cytoplasmic fraction of YAP and decreased the YAP nuclear fraction, but did not affect ERK1/2, Akt, and NF-κB activation in B16BL6 and 4T1 cells. These results indicate that RhoA/C promotes YAP activation in breast cancer and melanoma cells. Moreover, breast cancer and melanoma patients with high levels of YAP expression had significantly lower overall survival and recurrence-free survival rates than with patients with breast cancer and melanoma with low expression levels. It has been reported that higher expression of YAP is associated with a reduction in overall survival in colorectal carcinoma patients [51]. In addition, an enhanced YAP expression is a poor prognostic marker of survival in patients with pancreatic cancer [52]. These results suggest that the activation/overexpression of YAP is strongly correlated with poor prognosis in various cancers, including melanoma and breast cancer.

It has been shown that activation of YAP promotes tumor cell migration, invasion and adhesion [45,53]. In this study, we found that rhosin suppressed cell adhesion to type I and IV collagen via decreased RHAMM expression, but did not affect the expression of CD44 in B16BL6 and 4T1 cells. In addition, rhosin inhibited SDF-1-induced cell migration and invasion by suppressing CXCR4 expression. Moreover, treatment with rhosin abrogated the lung metastasis of B16BL6 and 4T1 cells in vivo. Cell membrane-localized RHAMM interacts with CD44 and can substitute for CD44 function [54]. It has been demonstrated that knock down of RHAMM decreases cell adhesion to ECM components, but not CD44 [45]. Overexpression of RHAMM is involved in colorectal carcinoma cell migration and invasion and leads to larger and fast-growing tumors. Downregulating its expression has been found to inhibit metastasis in xenograft models. Liver metastatic tumors have higher expression of RHAMM than the primary tumors in patients with colorectal carcinoma [55]. In addition, an increased RHAMM expression is significantly involved with a higher tumor grade and poor prognostic factors for overall and metastasis-free survival in breast cancer patients [56]. SDF-1 is advocated to fascinate CXCR4-expressing tumor cells to specific metastatic tissues, and high levels of SDF-1 are found in organs commonly affected by tumor metastasis, such as the lung [57,58]. It has also indicated that human pancreatic cancer cells with a high metastatic potential have been shown to upregulate the expression of CXCR4, but not CCR7, resulting in lung and liver metastasis in vivo, and a CXCR4 inhibitor suppressed the metastasis [57]. Moreover, patients with triple-negative breast cancer with high expression of CXCR4 had a poor prognosis compared to patients with low CXCR4 expression [59]. It has also been reported that mammary carcinoma and melanoma cell lines transfected with an activated mutant YAP promote tumor growth and metastasis in vivo [60]. RhoA accelerates YAP nuclear translocation, and its dominant negative form suppresses the YAP activation [61]. Activation of YAP induced cell motility via enhancing RHAMM expression [27]. Our results showed that verteporfin, a YAP inhibitor, suppressed the cell adhesion, migration and invasion via inhibition of RHAMM and CXCR4 expression in B16BL6, thereby the activation of the Rho/YAP pathway increased CXCR4 expression, thereby promoting cell migration and invasion. It has indicated that verteporfin suppressed the cell migration via inhibition of YAP nuclear translocation in Saos2 cells, patient-derived osteosarcoma, and GB-d1, a gallbladder cancer cell line [62,63]. In addition, inhibition of YAP-induced Slug expression by verteporfin suppressed the lung cancer cell migration and invasion via decreasing the epithelial-mesenchymal transition [64]. These findings suggest that inhibition of RhoA/C or YAP by Rho inhibitors, such as rhosin, suppresses tumor metastasis through the inhibition of RHAMM and CXCR4 expression in breast cancer and melanoma.

## 5. Conclusions

This study demonstrates that the RhoA or RhoC/YAP pathway accelerated tumor metastasis via increased expression of RHAMM and CXCR4 in melanoma and breast cancer cells. In addition, the inhibitors of rhosin suppressed RHAMM and CXCR4 expression, leading to inhibition of tumor metastasis. These results suggest that RhoA/RhoC and YAP could become promising targets for molecular therapy, and rhosin may provide an extremely useful pharmacotherapeutic approach in the treatment of melanoma and breast cancer patients.

## Figures and Tables

**Figure 1 biomedicines-09-00035-f001:**
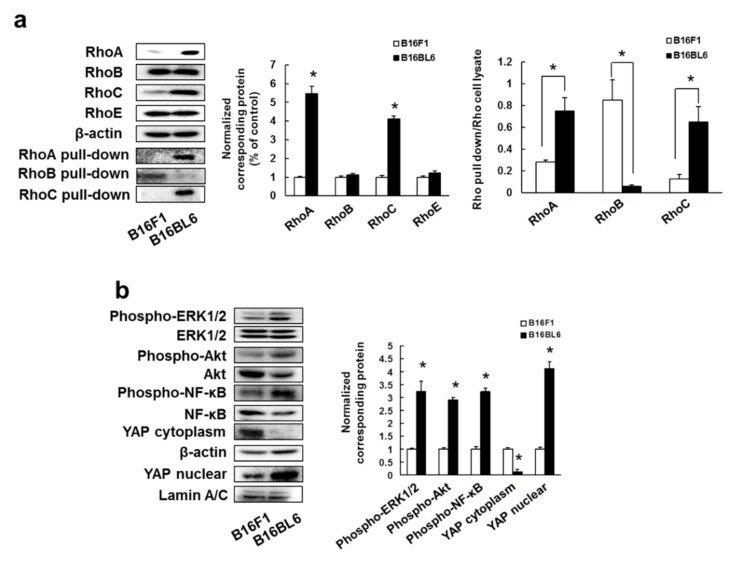
B16BL6 cells enhanced expression and activation of RhoA and RhoC: (**a**) Images of Western blots for the RhoA, RhoB, RhoC, RhoE, RhoA-GTP form (RhoA pull-down), RhoB-GTP form (RhoB pull-down), RhoC-GTP form (RhoC pull-down) and β-actin (internal standard), and quantification of the amounts of RhoA, RhoB, RhoC, RhoE, RhoA pull-down, RhoB pull-down, and RhoC pull-down after normalization to the amounts of corresponding protein. The results are representative of 4 independent experiments. * *p* < 0.01 vs. B16F1 cells (ANOVA with Dunnett’s test); (**b**) Images of Western blots for the phospho-ERK1/2, ERK1/2, phospho-Akt, Akt, phospho-NF-κB, NF-κB, YAP, β-actin, and lamin A/C, and quantification of the amounts of phospho-ERK1/2, phospho-Akt, phospho-NF-κB, and YAP after normalization to the amounts of corresponding protein. The results are representative of 4 independent experiments. * *p* < 0.01 vs. B16F1 cells (ANOVA with Dunnett’s test).

**Figure 2 biomedicines-09-00035-f002:**
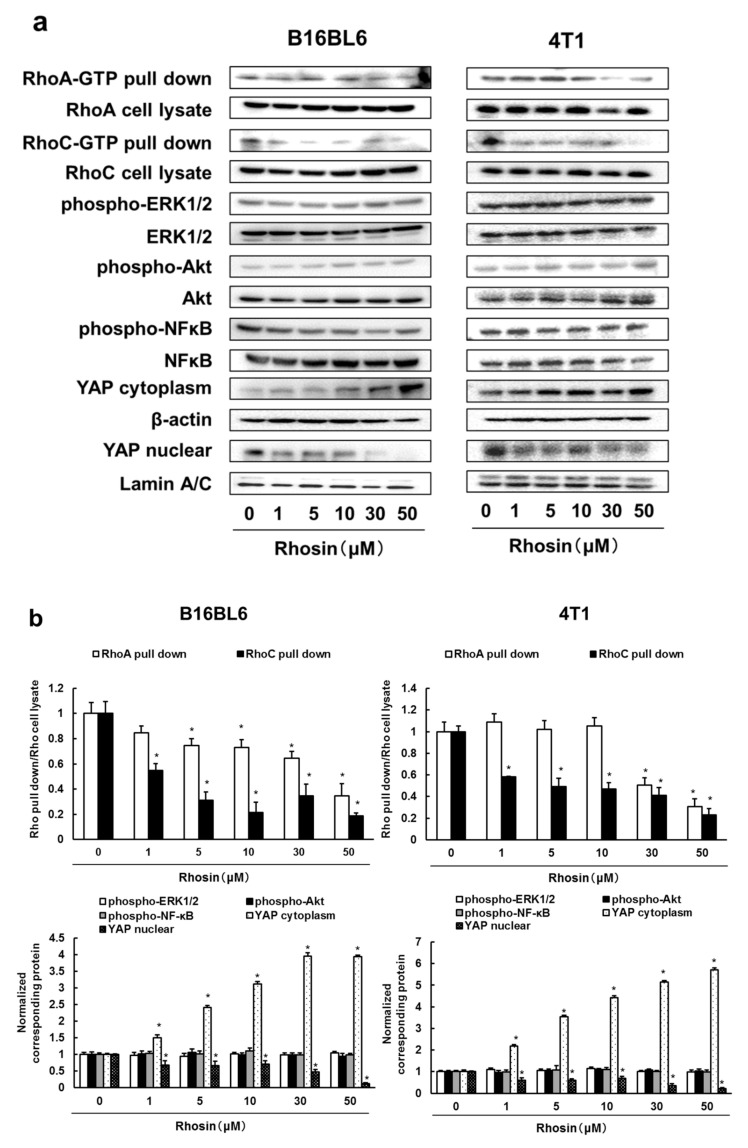
Rhosin inhibits RhoA/C-YAP pathway in B16BL6 and 4T1 cells: (**a**) B16BL6 and 4T1 cells were treated with rhosin at indicated concentration for 3 days. Images of Western blots for the RhoA pull-down, RhoA, RhoC pull-down, RhoC, phospho-ERK1/2, ERK1/2, phospho-Akt, Akt, phospho-NF-κB, NF-κB, YAP, β-actin, and lamin A/C; (**b**) Quantification of the amounts of RhoA pull-down, RhoC pull-down, phospho-ERK1/2, phospho-Akt, phospho-NF-κB, and YAP after normalization to the amounts of corresponding protein. The results are representative of 4 independent experiments. * *p* < 0.01 vs. controls (ANOVA with Dunnett’s test); (**c**,**d**) B16BL6 cells were treated with negative siRNA, (**c**) RhoA siRNA, or (**d**) RhoC siRNA. Images of Western blots for the RhoA, RhoC, YAP, β-actin, and lamin A/C. Quantification of the amounts of RhoA, RhoC, and YAP after normalization to the amounts of corresponding protein. The results are representative of 4 independent experiments. * *p* < 0.01 vs. controls (ANOVA with Dunnett’s test).

**Figure 3 biomedicines-09-00035-f003:**
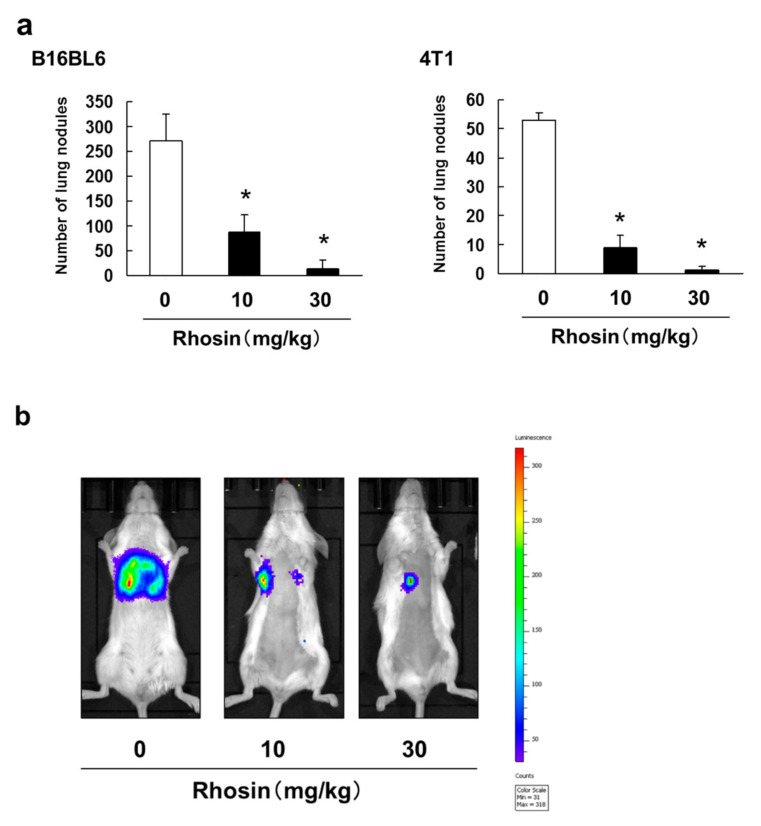
Inhibitory effect of intraperitoneal administration of rhosin on lung metastasis. (**a**) B16BL6 cells (1 × 10^5^ cells in 0.2 mL) and 4T1 cells (1 × 10^5^ cells in 0.2 mL) were injected into the tail vein of syngeneic C57BL/6J mice and Balb/c mice. Mice were treated daily from days 1 to 14 with 10 or 30 mg/kg rhosin. After 14 days, visible nodules that had metastasized to the lungs were counted. The results are expressed as the mean ± SD for 10 mice. * *p* < 0.01 vs. the controls (0.1% DMSO-treated) (ANOVA with Dunnett’s test); (**b**) Representative mice injected with 4T1-luc cells (1 × 10^5^ cells in 0.2 mL). Mice were treated daily from days 1 to 14 with 10 or 30 mg/kg rhosin. On day 14, tumor cells in mice were detected by IVIS Lumina XRMS Series III Imaging System.

**Figure 4 biomedicines-09-00035-f004:**
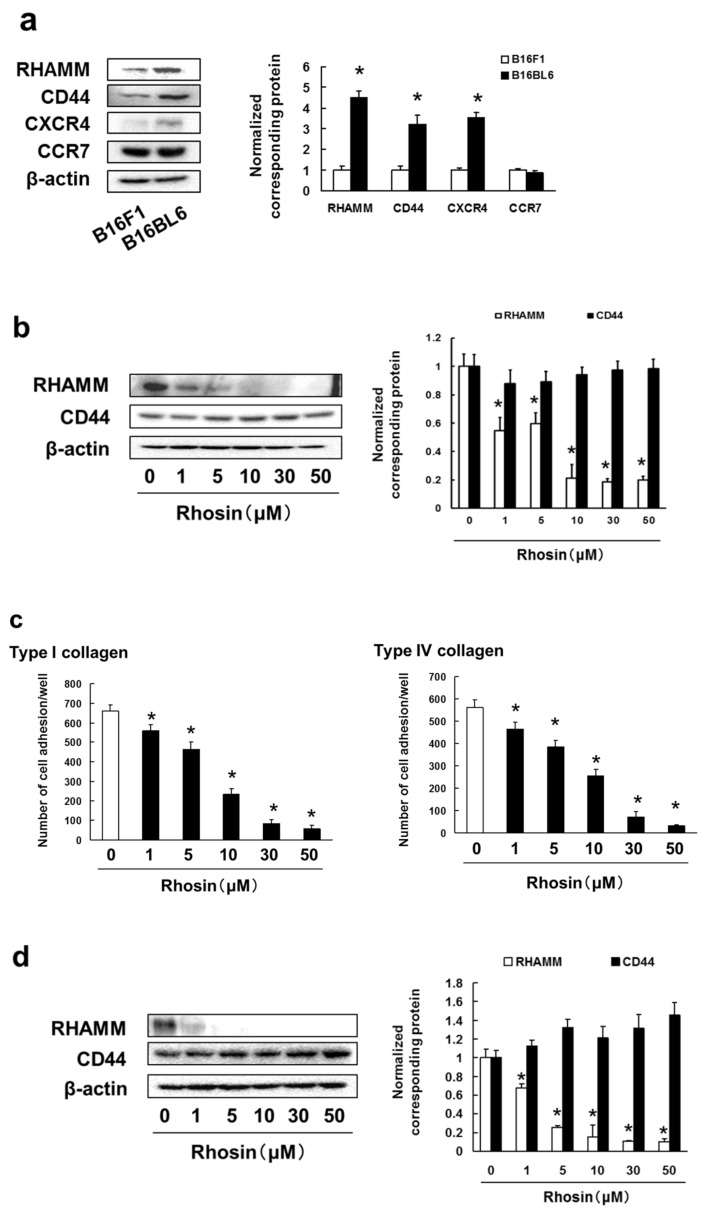
High metastatic potential cell lines of B16BL6 cells enhanced expression of RHAMM, CD44 and CXCR4: (**a**) Images of Western blots for the RHAMM, CD44, CXCR4, CCR7 and β-actin (internal standard), and quantification of the amounts of RHAMM, CD44, CXCR4 and CCR7 after normalization to the amounts of corresponding protein. The results are representative of 4 independent experiments. * *p* < 0.01 vs. B16F1 cells (ANOVA with Dunnett’s test); (**b**) B16BL6 cells were treated with rhosin at indicated concentration for 3 days. Images of Western blots for the RHAMM, CD44 and β-actin (internal standard), and quantification of the amounts of RHAMM and CD44 after normalization to the amounts of corresponding protein. The results are representative of 4 independent experiments. * *p* < 0.01 vs. controls (ANOVA with Dunnett’s test); (**c**) B16BL6 cells (1 × 10^4^ cells), which had been treated with rhosin for 3 days, were incubated with type I collagen- or type IV collagen-coated plates for 30 min at 37 °C in an atmosphere containing 5% CO_2_. The results are representative of 5 independent experiments. * *p* < 0.01, as compared to the controls (0.1% DMSO-treated) (ANOVA with Dunnett’s test); (**d**) 4T1 cells were treated with rhosin at indicated concentration for 3 days. Images of Western blots for the RHAMM, CD44 and β-actin (internal standard), and quantification of the amounts of RHAMM and CD44 after normalization to the amounts of corresponding protein. The results are representative of 4 independent experiments. * *p* < 0.01 vs. controls (ANOVA with Dunnett’s test); (**e**) 4T1 cells (1 × 10^4^ cells), which had been treated with rhosin for 3 days, were incubated with type I collagen- or type IV collagen-coated plates for 30 min at 37 °C in an atmosphere containing 5% CO_2_. The results are representative of 5 independent experiments. * *p* < 0.01, as compared to the controls (0.1% DMSO-treated) (ANOVA with Dunnett’s test); (**f**,**g**) B16BL6 cells were treated with negative siRNA, (**f**) RhoA siRNA, or (**g**) RhoC siRNA at indicated concentration for 3 days. Images of Western blots for the RHAMM and β-actin (internal standard), and quantification of the amounts of RHAMM after normalization to the amounts of β-actin. The results are representative of 4 independent experiments. * *p* < 0.01 vs. controls (ANOVA with Dunnett’s test); (**h**) B16BL6 cells (1 × 10^4^ cells), which had been treated with negative siRNA, RhoA siRNA, or RhoC siRNA for 3 days, were incubated with type I collagen- or type IV collagen-coated plates for 30 min at 37 °C in an atmosphere containing 5% CO_2_. The results are representative of 5 independent experiments. * *p* < 0.01, as compared to the controls (0.1% DMSO-treated) (ANOVA with Dunnett’s test).

**Figure 5 biomedicines-09-00035-f005:**
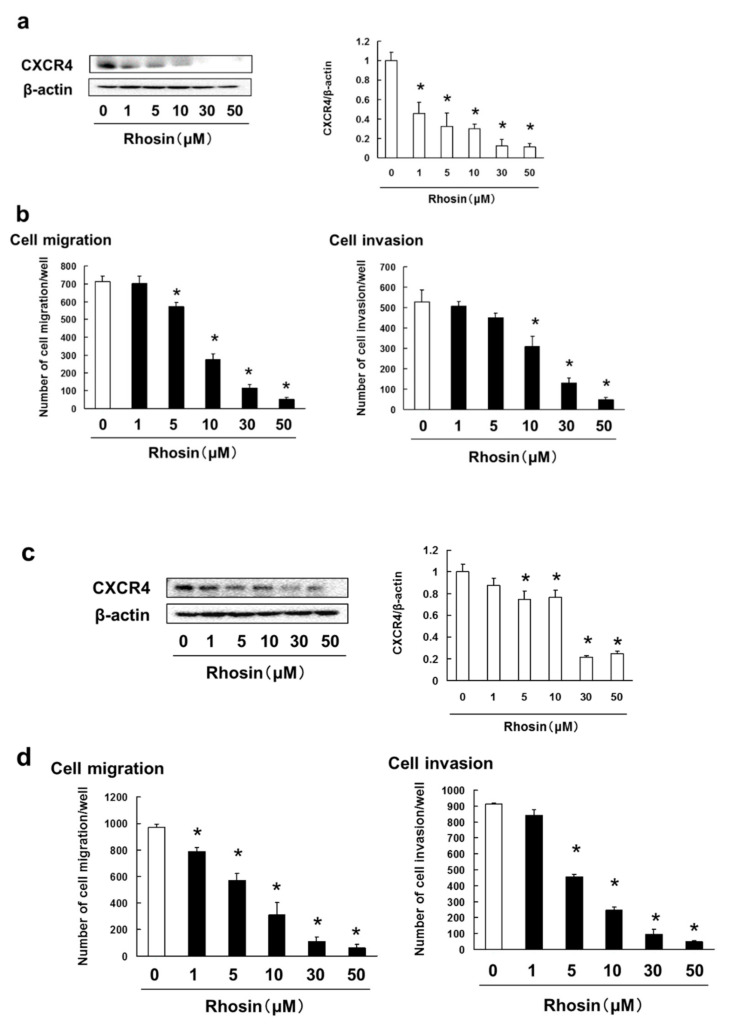
Rhosin suppressed SDF-1-induced cell migration and invasion via inhibition of CXCR4 expression: (**a**) B16BL6 cells were treated with rhosin at indicated concentration for 3 days. Images of Western blots for the CXCR4 and β-actin (internal standard), and quantification of the amounts of CXCR4 after normalization to the amounts of corresponding protein. The results are representative of 4 independent experiments. * *p* < 0.01 vs. controls (ANOVA with Dunnett’s test); (**b**) B16BL6 cells (1 × 10^4^ cells), which had been treated with rhosin for 3 days, were seeded into the upper compartments of chambers. Migration was analyzed by Boyden chamber assays using Falcon cell culture inserts. Invasive properties were analyzed using Falcon cell culture inserts covered with 50 µg of Matrigel per filter. For both assays, the lower chambers contained condition media including 50 ng/mL SDF-1 and 0.5% FBS. After incubation for 24 h, the cells invading the lower surface were counted microscopically. The results are representative of 5 independent experiments. * *p* < 0.01 vs. the controls (0.1% DMSO-treated) (ANOVA with Dunnett’s test); (**c**) 4T1 cells were treated with rhosin at indicated concentration for 3 days. Images of Western blots for the CXCR4 and β-actin (internal standard), and quantification of the amounts of CXCR4 after normalization to the amounts of corresponding protein. The results are representative of 4 independent experiments. * *p* < 0.01 vs. controls (ANOVA with Dunnett’s test); (**d**) 4T1 cells (1 × 10^4^ cells), which had been treated with rhosin for 3 days, were seeded into the upper compartments of chambers. Migration was analyzed by Boyden chamber assays using Falcon cell culture inserts. Invasive properties were analyzed using Falcon cell culture inserts covered with 50 µg of Matrigel per filter. For both assays, the lower chambers contained condition media including 50 ng/mL SDF-1 and 0.5% FBS. After incubation for 24 h, the cells invading the lower surface were counted microscopically. The results are representative of 5 independent experiments. * *p* < 0.01 vs. the controls (0.1% DMSO-treated) (ANOVA with Dunnett’s test); (**e,f**) B16BL6 cells were treated with negative siRNA, (**e**) RhoA siRNA, or (**f**) RhoC siRNA at indicated concentration for 3 days. Images of Western blots for the CXCR4 and β-actin (internal standard), and quantification of the amounts of CXCR4 after normalization to the amounts of β-actin. The results are representative of 4 independent experiments. * *p* < 0.01 vs. controls (ANOVA with Dunnett’s test); (**g**) B16BL6 cells (1 × 10^4^ cells), which had been treated with negative siRNA, RhoA siRNA, or RhoC siRNA for 3 days, were seeded into the upper compartments of chambers. Migration was analyzed by Boyden chamber assays using Falcon cell culture inserts. Invasive properties were analyzed using Falcon cell culture inserts covered with 50 µg of Matrigel per filter. For both assays, the lower chambers contained condition media including 50 ng/mL SDF-1 and 0.5% FBS. After incubation for 24 h, the cells invading the lower surface were counted microscopically. The results are representative of 5 independent experiments. * *p* < 0.01 vs. the controls (0.1% DMSO-treated) (ANOVA with Dunnett’s test).

**Figure 6 biomedicines-09-00035-f006:**
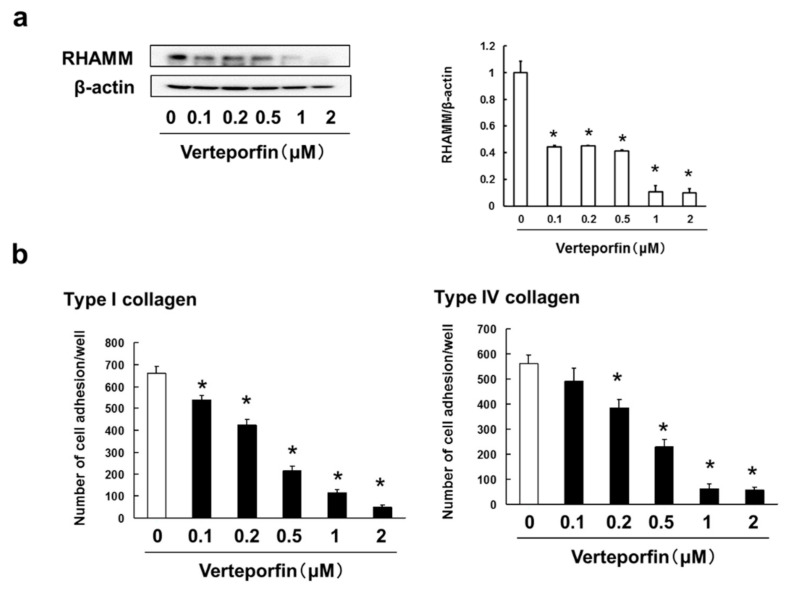
Verteporfin suppressed cell adhesion, and SDF-1-induced cell migration and invasion via inhibition of RHAMM and CXCR4 expression: (**a**) B16BL6 cells were treated with verteporfin at indicated concentration for 3 days. Images of Western blots for the RHAMM, CD44, and β-actin (internal standard), and quantification of the amounts of RHAMM and CD44 after normalization to the amounts of corresponding protein. The results are representative of 4 independent experiments. * *p* < 0.01 vs. controls (ANOVA with Dunnett’s test); (**b**) B16BL6 cells (1 × 10^4^ cells), which had been treated with verteporfin for 3 days, were incubated with type I collagen- or type IV collagen-coated plates for 30 min at 37 °C in an atmosphere containing 5% CO_2_. The results are representative of 5 independent experiments. * *p* < 0.01, as compared to the controls (0.1% DMSO-treated) (ANOVA with Dunnett’s test); (**c**) B16BL6 cells were treated with verteporfin at indicated concentration for 3 days. Images of Western blots for the CXCR4 and β-actin (internal standard), and quantification of the amounts of CXCR4 after normalization to the amounts of corresponding protein. The results are representative of 4 independent experiments. * *p* < 0.01 vs. controls (ANOVA with Dunnett’s test); (**d**) B16BL6 cells (1 × 10^4^ cells), which had been treated with verteporfin for 3 days, were seeded into the upper compartments of chambers. Migration was analyzed by Boyden chamber assays using Falcon cell culture inserts. Invasive properties were analyzed using Falcon cell culture inserts covered with 50 µg of Matrigel per filter. For both assays, the lower chambers contained condition media including 50 ng/mL SDF-1 and 0.5% FBS. After incubation for 24 h, the cells invading the lower surface were counted microscopically. The results are representative of 5 independent experiments. * *p* < 0.01 vs. the controls (0.1% DMSO-treated) (ANOVA with Dunnett’s test).

## Data Availability

The data presented in this study are available on request from the corresponding author.

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
