# Peer review of "Rhosin Suppressed Tumor Cell Metastasis through Inhibition of Rho/YAP Pathway and Expression of RHAMM and CXCR4 in Melanoma and Breast Cancer Cells"

_biomedicines, 2021, doi:10.3390/biomedicines9010035_

Round 1

Reviewer 1 Report

Tsubaki et al in their manuscript titled “Rhosin suppressed tumor cell metastasis through inhibition of Rho/YAP pathway and expression of RHAMM and CXCR4 in melanoma and breast cancer cells“ investigated the mechanism of metastasis in highly metastatic melanoma and breast cancer cells, and the mechanism of inhibition of metastasis by rhosin. The authors found that rhosin suppressed the RhoA and RhoC activation, the nuclear localization of YAP, but did not affect ERK1/2, Akt, or NF-κB activation in the highly metastatic cell lines B16BL6 and 4T1. High expression of YAP was associated with poor overall and recurrence free survival in patients with breast cancer or melanoma. Treatment with rhosin inhibited lung metastasis in vivo. Moreover, rhosin inhibited tumor cell adhesion to the extracellular matrix via suppression of RHAMM expression, and inhibited SDF-1-induced cell migration and invasion by decreasing CXCR4 expression in B16BL6 and 4T1 cells. These results suggest that the inhibition of RhoA/C-YAP pathway by rhosin could be an extremely useful therapeutic approach in patients with melanoma and breast cancer.

Major: The authors wrote that the tail vein injection leads to metastasis in 14 days is surprising. The authors should present some of the figures from the animal experiments to support their findings.

Minor: The authors should mention the cell lines and their respective cancers

The authors wrote that the for the effects of intraperitoneal administration of rhosin on lung metastasis of tumor cells 1 × 10 5 B16BL6 and 4T1 cells were administered into the tail vein of C57BL/6J mice and Balb/c mice. B16BL6- and 4T1-bearing mice were randomly divided into three groups, and were treated 112 with 0.1% dimethyl sulfoxide (control), and 10 mg/kg or 30 mg/kg of rhosin for 14 days. After 14 days, the mice were sacrificed, and the lung tissues of each mouse were excised. The tissues were ixed in a neutral-buffered formaldehyde solution, and the lung nodules were counted as black or white forms.

Did the mice develop metastasis within 14 days?

4T1 cells, a known to murine high metastatic breast cancer cell lines [29; 30], had express total and GTP-form RhoA and RhoC, and was potential of lung metastatic formation (Supplementary Fig. S2). How can the authors explain that breast cancer cells metastasize to lung

The authors presented data for B16BL6 and 4T1 cell lines for all the figures but only here and there for B16F1. What is the reason?

Reviewer 2 Report

In this manuscript the authors investigate the molecular mechanisms by which rhosin, a known RhoGTPases inhibitor, inhibits the metastasis of breast cancer cells as well as melanoma cancer cells. To this aim, the authors use two highly metastatic cell lines, B16BL6 and 4T1, as cell models for malignant melanoma and breast cancer respectively. They perform a comparative study of the expression and activation of different members of the RhoGTPase subfamily and downstream effectors in B16BL6 and B16F1, a melanoma cell line with low metastatic potential, to identify molecules responsible for the high metastatic potential of this type of cancer. The authors evaluate the effect of rhosin in these molecules as well as the effect of this compound in adhesion, migration and invasion in vitro and lung metastasis in vivo. Based on the results of these studies, the authors conclude that the activation of RhoA/C-YAP pathway contributes to tumor metastasis by increasing the expression of RHAMM and CXCR4, and that rhosin inhibits metastasis by blocking this pathway.

The study is interesting, provides valuable information for those working on cancer and has potential clinical relevance. However, there are some major points that should be addressed before considering the work for publication:

Major point:

  1. The experiments performed are insufficient to support the conclusions. Rescue experiments with active mutants should be included to further support that rhosin suppress metastasis through inhibition of Rho/YAP and expression of RHAMM and CXCR4.
  • Rescue of the nuclear localization of YAP after treatment with rhosin (Fig 2a) in cells transfected with constitutively active RhoA and/or RhoC, or with a mutant that is insensitive to rhosin (see Ref 23) should be included
  • Rescue of the expression of CXCR4 and RHAMM after treatment with rhosin (Figs. 4 and 5) in cells transfected with constitutively active form of YAP (see “The YAP1 Signaling Inhibitors, Verteporfin and CA3, Suppress the Mesothelioma Cancer Stem Cell Phenotype”. Kandasamy S. et al. Mol Cancer Res 2020 Mar; 18(3):343-351. doi: 10.1158/1541-7786.). This should further demonstrate that the effect of rhosin on CXCR4 and RHAMM expression is mediated through inhibition of YAP.

Additional points

  1. RhoB is highly expressed in both, B16F1 and B16BL6 (Fig 1a). Since RhoB is also a target of rhosin, a pull-down assay to evaluate the activation state of this GTPase in these cells should be included in Fig 1.
  2. The sentence “Rhosin is known to inhibit...” (lines 68-70) on introduction contains inaccurate data that need to be corrected:
  • Rhosin does not inhibit Rho GEFs, but the activation of Rho by GEFs.
  • Rhosin cannot inhibit cell invasion of MCF10A, since this cell line is not invasive per se. The reference 23, cited for this data, does not include this statement.
  1. The quantification of the active GTPases shown in the histograms of Fig 1a and 2b is not accurate. Active RhoGTPases have to be referred to the total amount of the corresponding GTPase (RhoA-GTP vs. total RhoA in each cell line and so on). This is important especially in results of Fig 1a, since there is much lower expression of RhoA and RhoC in B16F1 vs. the expression in B16BL6. Thus, this factor has to be taken into account to calculate the actual quantity of active RhoA and RhoC.
  2. Figs 1 and 2a. For clarity, it should be included the fraction used for the western-blot analysis of each proteins: cytosolic/nuclear. A more detailed description of the preparation of cell lysates should be included in Methods.
  3. The number of experiments performed for Fig 1 and 2b should be included.
  4. Include a description of the metastatic potential of B16F1 cells (low metastatic potential?).
  5. Results, especially those shown on Fig. 6 should be more extensively explained.
  6. The criteria for showing the results in the main text or in supplementary information are not clear. Since the conclusions of the paper apply to melanoma and breast cancer cells, results on breast cancer cells shown in supplementary should be moved to the main text (Fig. S4 should be shown in Fig. 3, S5 in Fig.4 and S6 in Fig.5). Results shown in Fig 2c-2d should be moved to supplementary. These results show the correlation between the levels of expression of YAP and prognosis of patients with melanoma. However, the paper focus on the activation of YAP by analysis of the subcellular localization, but no study about the levels of expression in cell lines is included.
  7. English usage needs attention. There are various typos and there are also sentences that are difficult to understand. As examples:
  • Line 98-99; The upper chambers were seeded into the cells... change for The cells were seeded into the upper chambers...
  • Sentences from line 140 to line 3142 (Moreover, 4T1 cells…) need to be rephrased.
  • Line 187. Inhibitory effect of rhosin on lung metastasis in B16BL6 and 4T1 cells. Change for Inhibitory effect of rhosin on lung metastasis in mice injected with B16BL6 and 4T1 cells.
  • Line 200. B16BL6 cells increased the expression... Change for B16BL6 cells have increased expression of ...
